# Getting to a feasible income equality

Ji-Won Park[1]*, Chae Un Kim[2]*

1 Regional Science, Cornell University, Ithaca, New York, United States of America, 2 Department of Physics, Ulsan National Institute of Science and Technology (UNIST), Ulsan, Korea

* jp429@cornell.edu (JWP); cukim@unist.ac.kr (CUK)

## Abstract

Income inequality is known to have negative impacts on an economic system, thus has been debated for a hundred years past or more. Numerous ideas have been proposed to quantify income inequality, and the Gini coefficient is a prevalent index. However, the concept of perfect equality in the Gini coefficient is rather idealistic and cannot provide realistic guidance on whether government interventions are needed to adjust income inequality. In this paper, we first propose the concept of a more realistic and 'feasible' income equality that maximizes total social welfare. Then we show that an optimal income distribution representing the feasible equality could be modeled using the sigmoid welfare function and the Boltzmann income distribution. Finally, we carry out an empirical analysis of four countries and demonstrate how optimal income distributions could be evaluated. Our results show that the feasible income equality could be used as a practical guideline for government policies and interventions.

**Data Availability Statement:** All relevant data are within the manuscript and its Supporting Information files.

**Funding:** Funder: The Samsung Science and Technology Foundation; Award Number: SSTF-BA1702-04; Grant Recipient: Chae Un Kim.

## Introduction

Income inequality is an enduring focus of inquiry in the social sciences [1–15]. Many scholars have addressed the negative impacts of income inequality [10, 16–18]. For example, Wilkinson and Pickett (2009) argued that countries with severe inequality have more problems connected with child well-being, drug abuse, education, imprisonment, obesity, physical and mental health, social mobility, teenage pregnancies, and violence.

Various ways have been proposed to quantitatively measure income inequality such as Atkinson's index, Gini coefficient, Hoover index, Theil index, and generalized entropy index [1, 19–28]. Among them, the Gini coefficient is a prevalent index, representing income inequality within a nation or any other group of people with a single number [23, 29, 30]. The Gini coefficient can be used to show whether the present income distribution is made more equal than it was in the past or whether less-developed countries are characterized by greater inequality than developed countries. Besides, the Gini coefficient shows if government interventions such as taxes can lead to greater equality in the income or wealth distribution.

The Gini coefficient ranges between 0 (representing perfect equality) and 1 (perfect inequality). Here, perfect equality can be achieved when everyone in a given nation or society has the same level of income. It is frequently alleged within the political debate that forcing this perfect equality leads to an economic inefficacy thus to a less productive society. It is

**Competing interests:** The authors have declared that no competing interests exist.

believed that perfect income equality is achievable only when everyone is identical and has equal capability in economic contributions. In other words, to achieve perfect income equality, factors that affect individual incomes, such as intelligence, inherited health and wealth, personalities such as persistence and confidence, and social skills need to be identical for everyone. In reality, however, these conditions cannot be met. Additionally, it is well-known that the income incentives for talented people can lead to greater wealth in society overall [31]. Therefore, it is evident that the concept of perfect income equality is rather idealistic and practically infeasible in the real world.

If a more realistic and feasible concept of income equality can be implemented into the Gini coefficient, it can serve as a practical guideline for a realistic and practical income distribution, ensuring the maximization of overall social welfare without hampering the overall economic efficacy. In this paper, we propose a feasible equality line that can be incorporated into the Gini coefficient. We first describe the basic concept of a feasible equality line and then develop its mathematical formula using the sigmoid welfare function and the Boltzmann distribution. Then, through empirical data analysis of four countries, we show how feasible equality lines can be used in practice to guide government policies and interventions.

## Model development

### Lorenz curve and the concept of feasible equality line

When calculating the Gini coefficient, the Lorenz curve is needed. The Lorenz curve, developed by Max O. Lorenz in 1905, is a prevailing way of displaying the distribution of income within an economy during a given year [32]. The Lorenz curve plots the proportion of the total income of the population (y-axis) that is cumulatively earned by the bottom x% of the population (Fig 1). The line with the slope of 45 degrees has been considered the ideal and perfect equality line of income. The more the Lorenz curve line is away from the perfect equality line, it is considered that the higher the degree of inequality is represented.

The perfect equality line can be achieved via a uniform income distribution among the population. However, as discussed earlier, this perfect equality line is idealistic and practically infeasible. In reality, there are no countries exhibiting perfect equality in their income distributions, and the Lorenz curves for the real-world countries lie somewhere to the right of the diagonal [33]. For example, the Lorenz curve of a typical nation is plotted in Fig 1. The Lorenz curve representing the national income distribution is located at the right of the perfect equality line. As the perfect equality line cannot serve as a reference line for the real world, a more practical and feasible equality line, such as the hypothetical feasible equality line in Fig 1, is required. If the actual Lorenz curve of a nation is located close to the feasible equality line, then the nation's income distribution can be considered practically reasonable and close to equal. On the other hand, if the Lorenz curve is located away from the feasible equality line, the nation's income distribution can be considered away from income equality. Therefore, the feasible equality line provides a useful reference guide for the government policies (for example, income taxes) for the redistribution of incomes. In the following section, we will develop a model to formulate a feasible equality line.

### Feasible income equality and sigmoid individual welfare function

In this study, we define feasible income equality as an optimal income distribution that maximizes total social welfare without hampering the sustainable economic growth of a given society. In the feasible income equality, income must be fairly distributed to individuals by properly reflecting the realistic factors affecting their economic contributions.

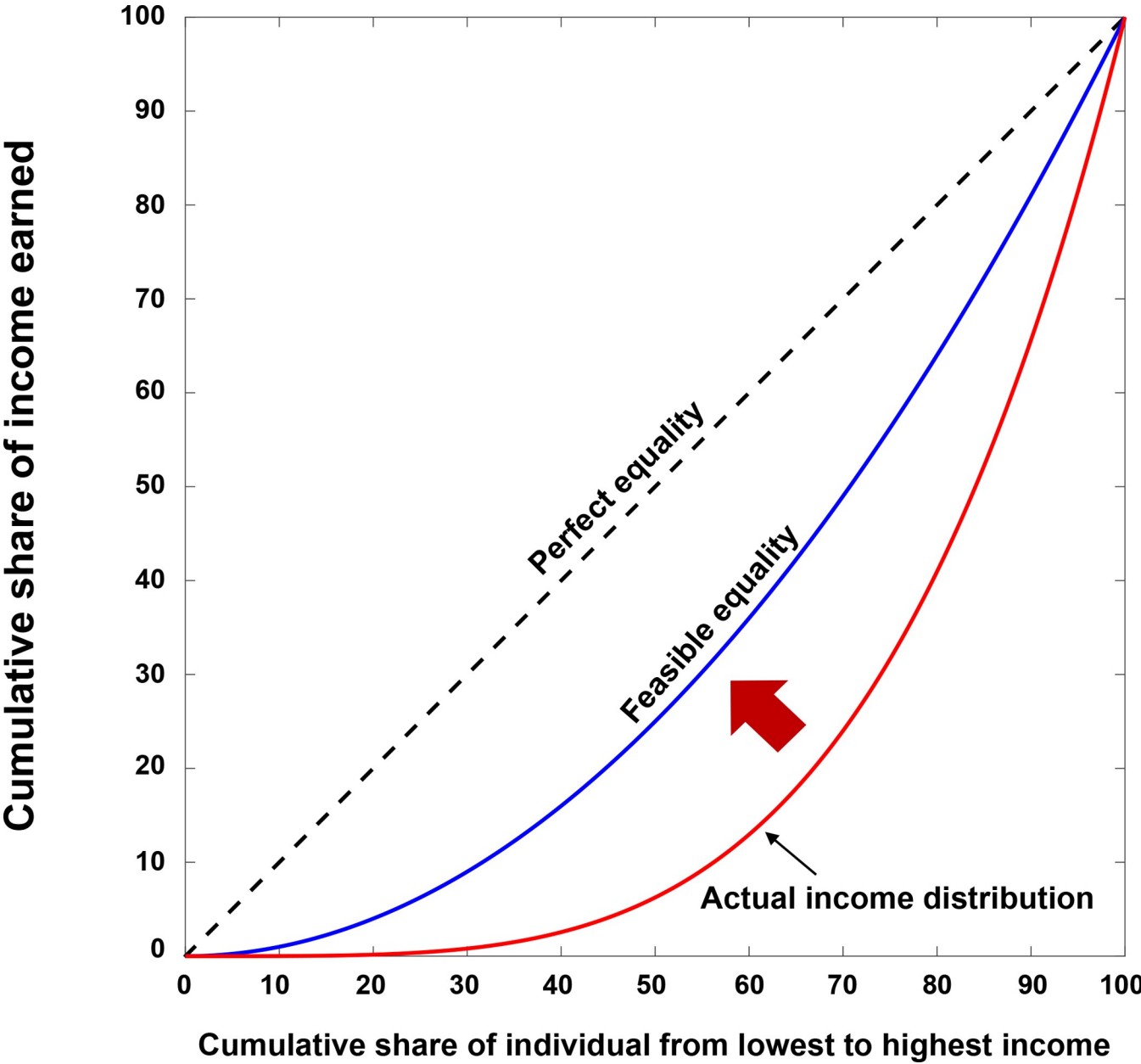

**Fig 1. The Lorenz curve of a typical country.** The hypothetical feasible equality line (blue) can serve as a practical guideline for government policies and interventions (red arrow).

In the past decades, several different kinds of social welfare functions have been suggested to quantitatively describe social welfare [1, 27, 34–36], and one of the simplest forms is the linear sum of all individual's welfare function (Eq 1).

$$W = \sum_{i=1}^{n} U(y_i), \ \ for \ i = 1, 2, \ldots, n \tag{1}$$

where $W$ is the total social welfare of a given society, $U(y_i)$ is the individual welfare function of individual $i$, and $n$ is the total population.

For example, the well-known Utilitarian social welfare function is expressed as Eq 1, in which the individual welfare function is defined as a linear function of an individual's income (Fig 2A). Although its form is straightforward, the Utilitarian social welfare function has an explicit limitation: it is entirely independent of the income distribution among the population and is only dependent on the total income sum of the society [19]. In other words, the individuals' economic contributions cannot be appropriately incorporated into the Utilitarian social welfare function. In this regard, the Utilitarian function cannot be a suitable target social welfare function for finding feasible income equality.

Considering that the independence of income distribution in the Utilitarian social welfare function originates from the linearity in the individual welfare function, a more realistic social welfare function can be obtained by replacing the linear individual welfare function in the Utilitarian social welfare with a non-linear individual welfare function. It is then clear that finding a proper non-linear individual welfare function is the key to the realistic social welfare function and thus for finding feasible income equality.

The non-linear individual welfare function should reflect realistic welfare (well-being, happiness, and satisfaction) that an individual feels as income increases. When the individual's income is close to zero, the welfare value must be at the minimum (or set to be 0%). The welfare value would increase as income increases, but not rapidly below the critical low-income value (such as minimal cost of living). This slow welfare increase is because the income is still insufficient to support basic living. When the income increases beyond the critical low-income value, the individual begins to have some degrees of economic freedom, therefore, the welfare value would increase suddenly rapidly. As the income increases further, the degrees of economic freedoms increase, but eventually become saturated at a critical high enough income value. At the critical high-income value, the welfare value would also be saturated, and afterward, the welfare value would increase rather slowly.

The non-linear behavior of the individual welfare described above can be represented by a sigmoid function with two constants, $\mu$ and $\alpha$ (Eq 2 and Fig 2B). A sigmoid function is an "S-shaped" non-linear function, which is used in a wide range of research fields, such as Fermi-Dirac distribution in Physics [37], and logistic function and welfare function for resource consumption in Economics [38–40]. The sigmoid welfare function is monotonically increasing from 0 (full unhappiness or dissatisfaction) to 1 (full happiness or satisfaction). At the income value $\mu$, the sigmoid welfare value becomes 0.5 and increases at the maximum rate (i.e., the first derivative with respect to income is at the maximum). The income value $\mu$ can be interpreted as the average value of the critical low-income value (point $L$ in Fig 2B) and the critical high-income value (point $H$ in Fig 2B). The other constant $\alpha$ determines the width between the $L$ and $H$ points. The width becomes narrower as $\alpha$ increases.

$$U(y) = \frac{1}{(1 + e^{\alpha(\mu-y)})} \tag{2}$$

Where $y$ is the income of an individual.

Then, the total social welfare function ($W$) can be defined as the sum of the sigmoid individual welfare functions.

$$W(y_1, y_2, \cdots, y_n) = \sum_{i=1}^{n} U(y_i) = \sum_{i=1}^{n} \frac{1}{(1 + e^{\alpha(\mu-y_i)})}, \; for \; i = 1, 2, \ldots, n \tag{3}$$

where $\alpha$ and $\mu$ are positive constants and $y_i$ is the income of individual $i$ in a given society.

Our goal is to find an optimal income distribution $\{y_1, y_2, \cdots y_n\}$ that maximizes the total social welfare function.

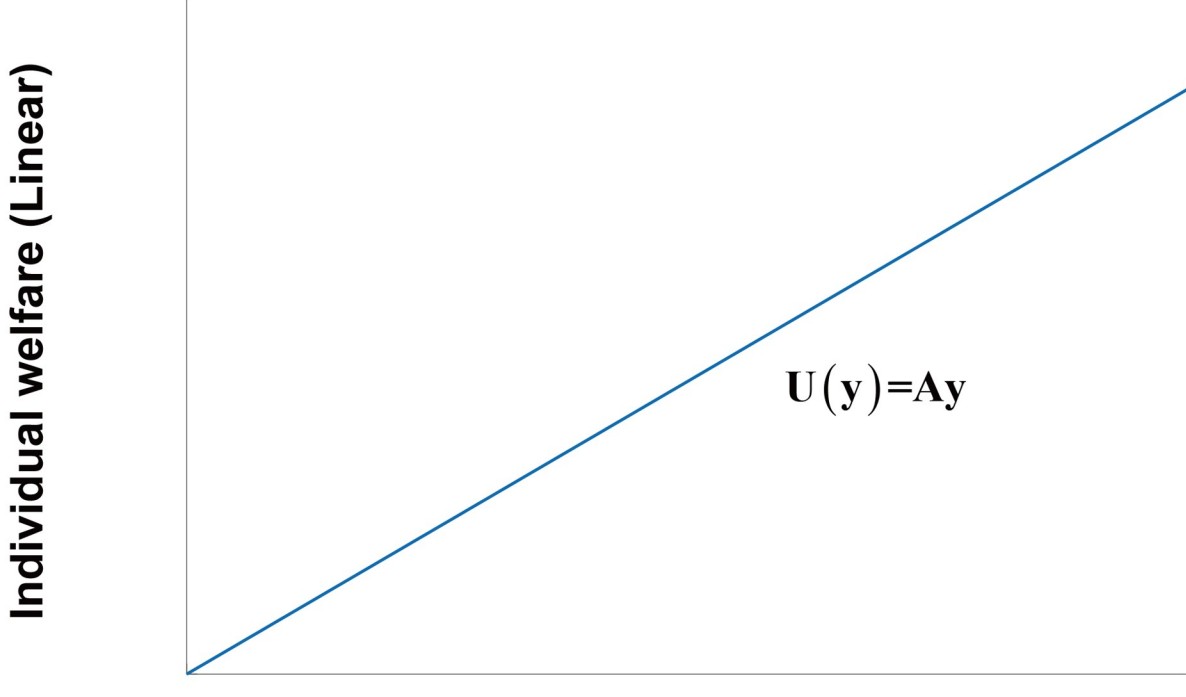

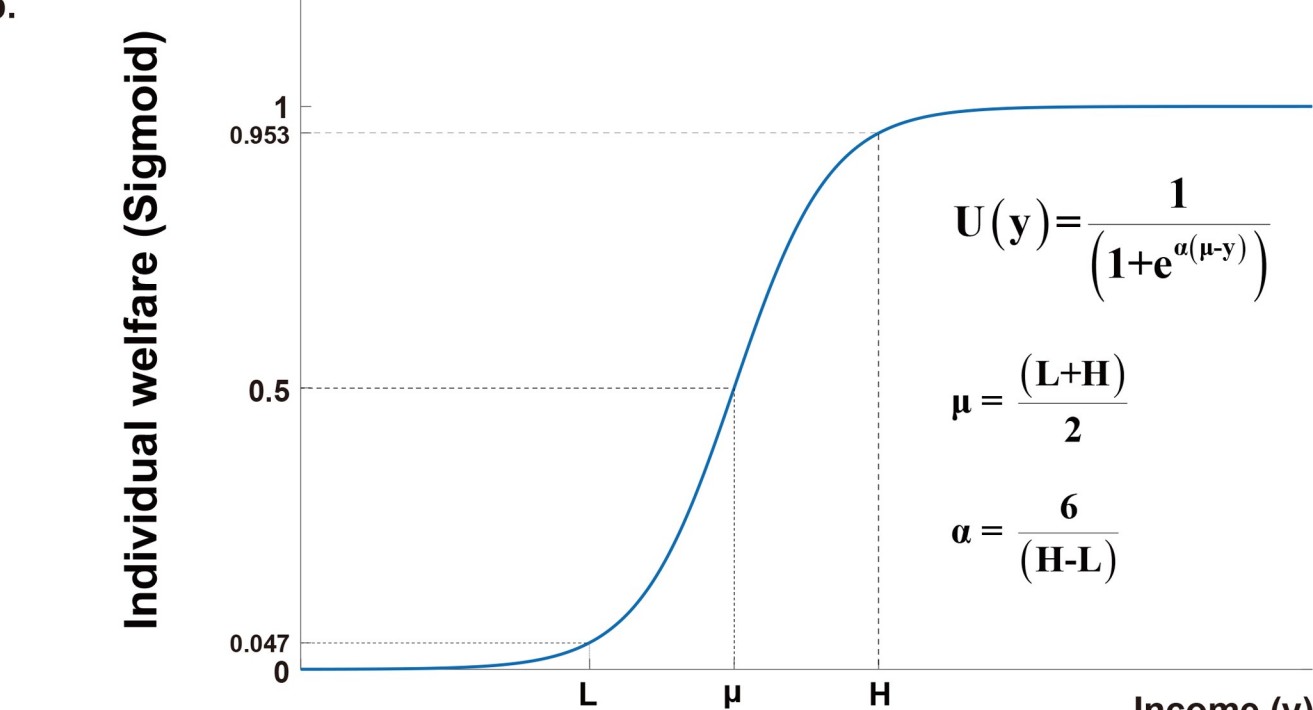

**Fig 2. Individual welfare functions as a function of income.** (a) The linear function used in the Utilitarian social welfare. (b) The non-linear sigmoid function, reflecting more realistic individual welfare as income increases. With the critical low- and high- income values ($L$ and $H$), the two constants ($\mu$ and $\alpha$) in the sigmoid function can be determined.

## Optimal income distribution using the Boltzmann distribution

The severe income inequality that we currently face is not simply due to the positive correlation between the individual's economic contributions and income. The problem instead originates from the fact that a few talented people earn too much income, whereas the others earn less than what they need or deserve. Therefore, if a given society's total income is distributed more fairly to the economic participants (individual 1,2,· · ·,$n$), the income should be distributed to the individuals in an unbiased manner. Such an unbiased (or fair) distribution of income can be achieved using the Boltzmann distribution [41].

In the physical sciences, the Boltzmann distribution yields the equilibrium probability distribution of a physical system in its energy substates [37, 42]. The description is valid in a classical physics regime in which each physical particle of the system is identical to but distinguishable from the others, and the interaction among the particles is negligible. In the Boltzmann distribution, the probability ($P_i$) that a particle can be found in the $i^{th}$ substate is inversely proportional to the exponential function of the substate energy ($E_i$) (i.e., $P_i \propto e^{-\beta E_i}$, $\beta = 1/kT$ ($k$ = Boltzmann constant, $T$ = absolute temperature)). The Boltzmann distribution, is based on entropy maximization and provides the most probable, natural, and unbiased distribution of a physical system at thermal equilibrium.

Over the past decades, various types of entropy-based approaches have been applied to social studies [43–49]. Atkinson (1970) proposed income inequality measurement using entropy, and Banerjee and Yakovenko (2010) presented that the money, income, and the global energy consumption distributions correspond to the entropy maximization for the partitioning of a limited resource among multiple agents. Banerjee and Yakovenko (2010) also showed that social and economic inequality is ubiquitous in the real world using the entropy maximization.

Unlike other studies, Park et al. (2012) applied entropy maximization in a different direction. Park et al. (2012) introduced entropy maximization to the problem of permit allocation in emissions trading. In their study, the concepts in a physical system were replaced by the concepts in an emissions trading system: the physical particle was replaced by the unit emissions permit, the physical substates by the individuals of the participating countries, and the potential energy $E_i$ of a physical substate $i$ by the allocation potential energy ($E_i$) of an individual in the country $i$. Then the probability that a unit emissions permit is allocated to a country $i$ became proportional to its total population ($C_i$) and was inversely proportional to the exponential function of the allocation potential energy $E_i$ (i.e. $P_i \propto C_i e^{-\beta E_i}$, $\beta$ is a positive constant). It was argued that the Boltzmann distribution in the initial permit allocation provides the most probable allocation among multiple countries. In addition, it was proposed that the concept of '*most probable*' in the physical sciences might be translated into '*fair*' in social sciences, as the distribution provides a natural and undistorted allocation among participants.

Inspired by the fairness concept brought to the social sciences, we now apply the approach using the Boltzmann distribution in Park et al., 2012 to the income distribution. The concepts in a physical system are replaced by the concepts in an income distribution system: the physical particle is replaced by the unit income, the physical substates by the individuals in a country, and the potential energy $E_i$ of a physical substate $i$ by the negative value of the *income distribution factor* of an individual $i$. The income distribution factor ($\tilde{E}_i$) of an individual $i$ is a measure of the economic contribution that could be made from combinations of various factors such as intelligence, personalities, and physical and social skills of the individual. Based on this definition, individuals with higher income distribution factors make higher economic contributions, therefore, deserve higher income. In addition, individuals with higher talents can have higher income distribution factors, therefore, tend to earn a higher income. Using

the income distribution factor, the comprehensive impact of the individual's economic contributions can be quantitatively incorporated into the income distribution.

In the Boltzmann income distribution, when the total income ($Y$) is distributed to $n$ individuals in a country, the probability ($P_i$) that a unit income is distributed to an individual $i$ can be expressed as the following.

$$P_i = \frac{e^{\beta \tilde{E}_i}}{\sum\limits_{i=1}^{n} e^{\beta \tilde{E}_i}}, \quad for \; i = 1, 2, \ldots, n \tag{4}$$

Where $\tilde{E}_i$ is the income distribution factor of individual $i$, and $\beta$ is a positive constant.

Then the income ($y_i$) distributed to an individual $i$ is

$$y_i = Y \times P_i, \quad for \; i = 1, 2, \ldots, n \tag{5}$$

Where $Y$ is the total income.

The obtained income distribution $\{y_i\}$ is simple with a single adjustable constant $\beta$, yet highly versatile. If $\beta$ approaches 0, all individuals receive an equal amount of income, representing uniform income distribution. If $\beta$ increases to a large value, then the probability ($P_i$) becomes non-zero values only for the few individuals with the highest income distribution factors, representing highly non-uniform income distribution. Thus, the income distribution using the Boltzmann distribution can represent a wide range of income distributions covering from the idealistic perfect equality to the perfect inequality.

When the income distribution $\{y_i\}$ based on the Boltzmann distribution is inserted into the total social welfare function (Eq 3), the total social welfare function becomes a function of the $\beta$ value. As shown in Fig 3, if the social welfare function can be maximized at a specific $\beta$ value (denoted by $\beta^*$), then the corresponding income distribution with the $\beta^*$ value represents the optimal income distribution.

Therefore, we search for the $\beta$ value that satisfies the following.

$$\underset{\beta}{Max} \;\; W(y_1, y_2, \ldots, y_n) = \sum_{i=1}^{n} \frac{1}{(1 + e^{\alpha(\mu - y_i)})} \tag{6}$$

subject to

$$y_i = Y \times \frac{e^{\beta \tilde{E}_i}}{\sum\limits_{i=1}^{n} e^{\beta \tilde{E}_i}}, \quad for \; i = 1, 2, \ldots, n \tag{7}$$

The first-necessary condition will be

$$\frac{\partial W}{\partial \beta} = \frac{\partial W}{\partial y_1} \cdot \frac{\partial y_1}{\partial \beta} + \frac{\partial W}{\partial y_2} \cdot \frac{\partial y_2}{\partial \beta} + \ldots + \frac{\partial W}{\partial y_n} \cdot \frac{\partial y_n}{\partial \beta} = 0 \tag{8}$$

## Empirical data analysis

To demonstrate the optimal income distribution representing the feasible income equality, we performed empirical analysis on the four selected countries: the U.S.A., China, Finland, and South Africa (Table 1). The U.S.A. and China are the two largest economies in the world in terms of GDP values. Finland is considered one of the most equal countries, and its Gini coefficient is lower (0.27 in 2017) than the others. In contrast to Finland, South Africa is considered one of the least equal countries in the world, with the highest Gini coefficient (0.63 in 2014).

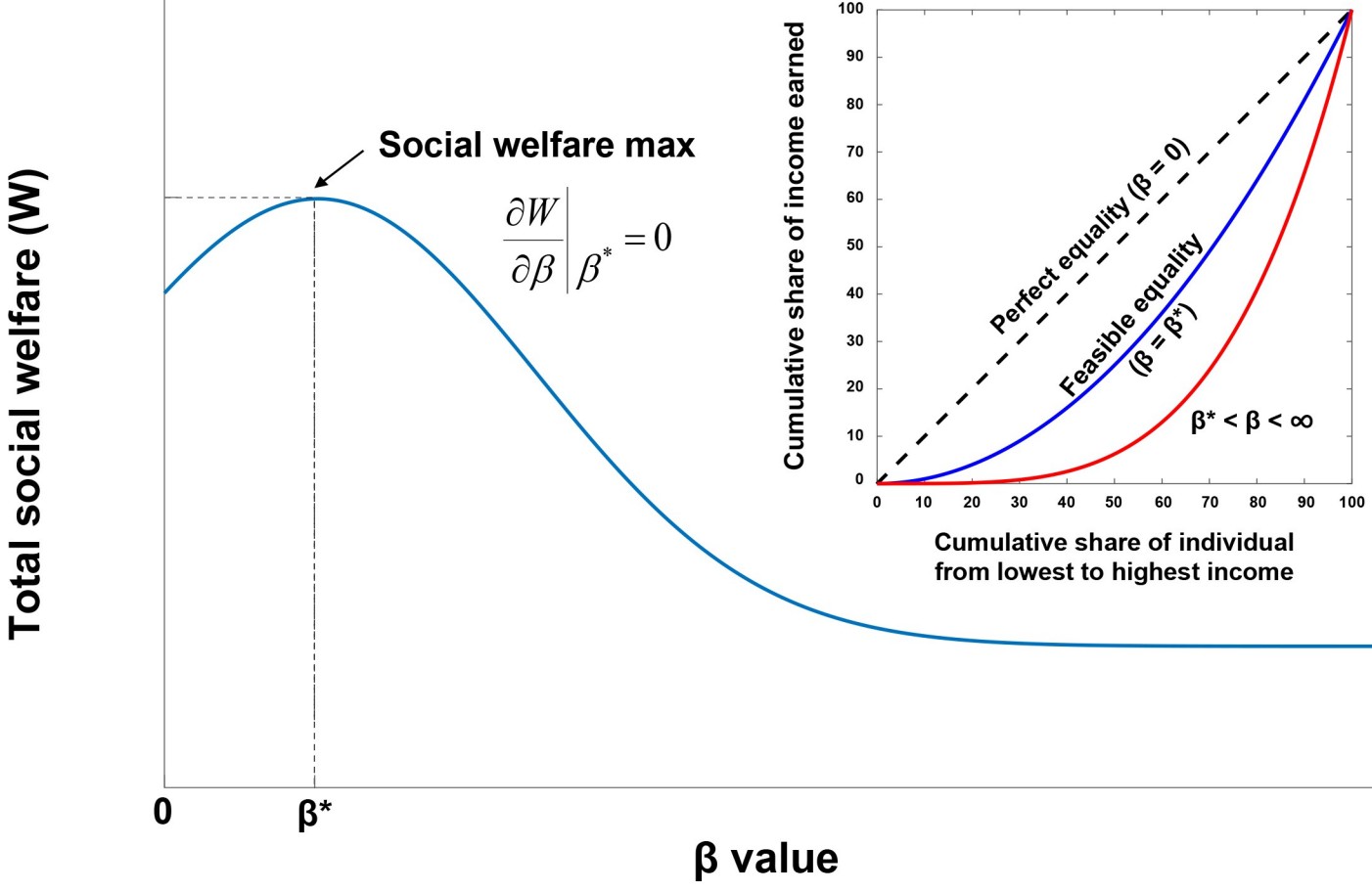

**Fig 3. Total social welfare as a function of $\beta$ value.** The welfare function is maximized at $\beta = \beta^*$, and the corresponding Lorenz curve (blue) represents the feasible equality line (the optimal income distribution).

In the empirical data analysis, we used the household income dispersion in the four countries (Table 1). Each country is divided into five subgroups (lowest, second, third, fourth, highest quintiles), and each subgroup has 20% household population of the country. The percentage share of the income is the share that accrues to the subgroups in the countries. To simplify data analysis, it was assumed in this study that all individual households in each subgroup have an equal amount of income. Then individual households in each country can have five different levels of income, and their income values are directly proportional to the

**Table 1. Share of household income in four countries.**

| Country (Year) | Lowest Quintile | Second Quintile | Third Quintile | Fourth Quintile | Highest Quintile | Gini (Year) | GDP (billion US$) (Year) |
|---|---|---|---|---|---|---|---|
| U.S.A.* (2019) | 3.1 | 8.3 | 14.1 | 22.7 | 51.9 | 0.484 (2019) | 21,374 (2019) |
| China (2016) | 6.5 | 10.7 | 15.3 | 22.2 | 45.3 | 0.385 (2015) | 14,343 (2019) |
| Finland (2017) | 9.4 | 14 | 17.4 | 22.3 | 36.9 | 0.274 (2017) | 269 (2019) |
| South Africa (2014) | 2.4 | 4.8 | 8.2 | 16.5 | 68.2 | 0.63 (2014) | 351 (2019) |

Source: World Development Indicators

https://data.worldbank.org/indicator/SI.DST.04TH.20

*Source: U.S. Census Bureau, Current Population Survey, 1968 to 2020 Annual Social and Economic Supplements (CPS ASEC).

percentage share values in Table 1. In the calculation, the percentage share values in Table 1 were regarded as the relative income values of individual households.

The relative income values of individual households were then used to determine the two constants, $\mu$ and $\alpha$, in the sigmoid individual welfare function (Eq 2). In this paper, we assumed that the critical low-income value ($L$) is located between the relative income values of the second and third quintiles and the critical high-income value ($H$) between the relative income values of the fourth and highest quintiles. The constants $\mu$ and $\alpha$ were defined as $\mu = (L + H)/2$, and $\alpha = 6/(H − L)$. With these parameter values, the sigmoid individual welfare function $U(y_i)$ has the following properties: $U(L) = 1/(1 + e^3) \sim 0.047$, $U(\mu) = 0.5$, $U(H) = 1/(1 + e^{-3}) \sim 0.953$ (Fig 2B). The $L$, $H$, $\mu$ and $\alpha$ values determined for the four countries are summarized in Table 2.

With the $\mu$ and $\alpha$ values determined, the total welfare function (Eq 6) becomes a function of the $\beta$ value in the Boltzmann distribution (Eq 7). We assumed in this study that the income distribution factor ($\tilde{E}_i$), which represents the economic contribution of individual household $i$, is directly proportional to the individual household's relative income value. Using this simple definition, a unit income is more likely to be distributed to an individual in the higher income subgroups in the Boltzmann distribution.

Fig 4 shows the total social welfare of the four countries as a function of $\beta$ value. In all cases, the total social welfare increases rapidly up to an optimal $\beta$ value (denoted as $\beta^*$), and then gradually decreases. It is interesting to note that the total social welfare at $\beta = 0$ is higher than when $\beta$ value is a large value. This result suggests that, in all countries, the total social welfare is higher with the completely uniform income distribution rather than with the significantly non-uniform income distribution, in which only the highest income subgroup shares most income.

The optimal income distributions corresponding to the social welfare maximization are summarized in Table 3. Compared to the optimal income distributions, the actual income distributions show deficiency up to the third quintile in the U.S.A, China, and Finland, and even to the fourth quintile in South Africa (Fig 5). Besides, the highest quintile in South Africa received significantly more income than the optimal income, suggesting the most biased actual income distribution (Fig 5D).

Fig 6 shows the Lorenz curves for the actual and the optimal income distributions of the four countries. In all countries, the Lorenz curve for the optimal income distribution, or the feasible income equality line, is located between the diagonal (idealistic perfect equal) line and the actual income line. It is also noticed that the feasible equality line has a similar line shape for all four countries. This observation is manifested by the Gini coefficient calculations (Table 3). The Gini coefficients for the actual income distributions are relatively widely distributed from 0.25 (Finland) to 0.57 (South Africa). On the other hand, the Gini coefficients for the optimal income distributions are narrowly distributed from 0.12 (Finland) to 0.17 (the U.S.A.). This result raises the possibility that a universal feasible equality line could be found and applicable to all countries in the world.

**Table 2. Parameter values for the individual social welfare functions.**

| Country | Critical low-income value | Critical high-income value | $\mu$ | $\alpha$ |
|---|---|---|---|---|
| | $L = \frac{(2nd\ Q + 3rd\ Q)}{2}$ | $H = \frac{(4th\ Q + Highest\ Q)}{2}$ | $\mu = \frac{(L+H)}{2}$ | $\alpha = \frac{6}{(H-L)}$ |
| U.S.A. | 11.20 | 37.30 | 24.25 | 0.23 |
| China | 13.00 | 33.75 | 23.38 | 0.29 |
| Finland | 15.70 | 29.60 | 22.65 | 0.43 |
| South Africa | 6.50 | 42.35 | 24.43 | 0.17 |

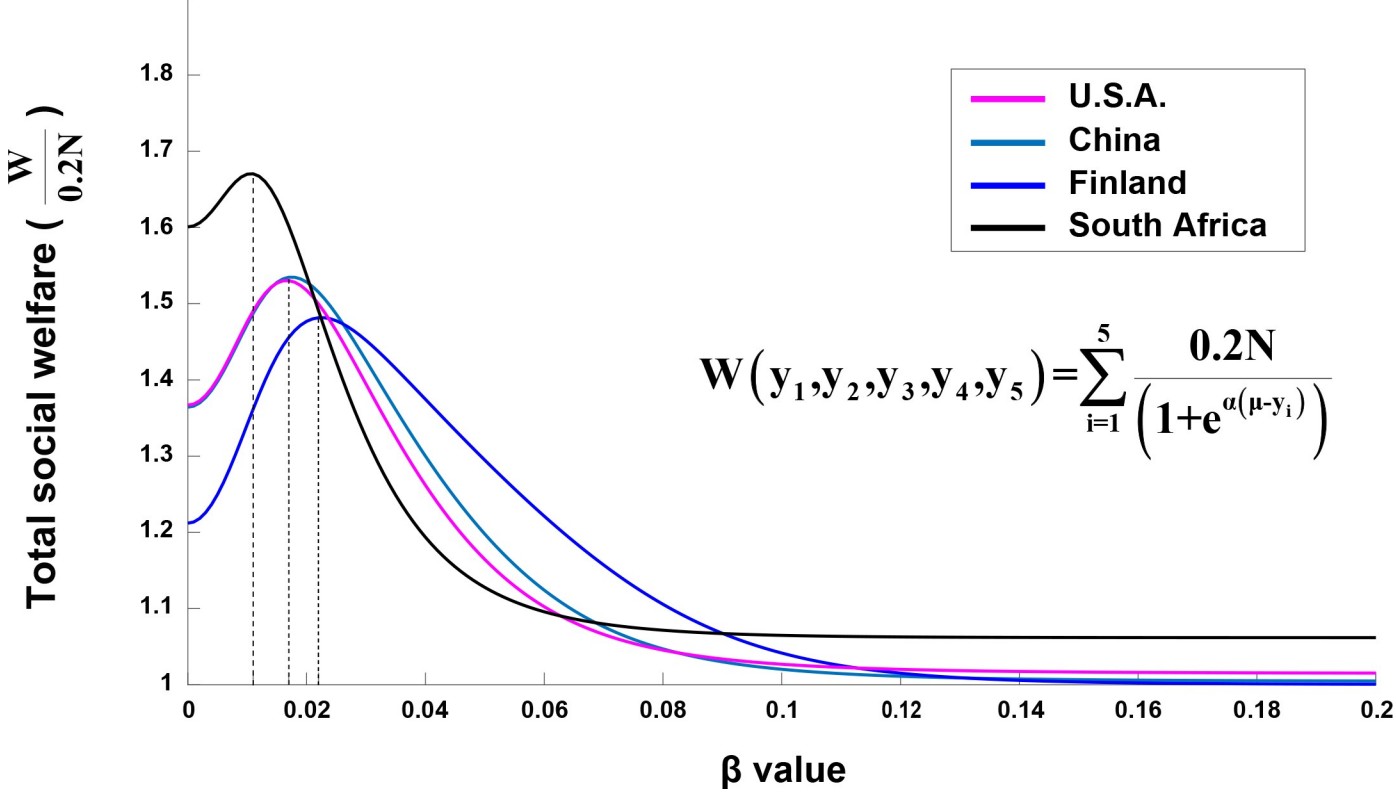

**Fig 4. Total social welfare as a function of *β* value in the four countries.** The social welfare plots are normalized to the number of individual households in each subgroup (0.2**N**, where **N** is the total number of individual households in a country). The total social welfare functions of the four countries are maximized at $\beta^*_{U.S.A.} = 0.017$, $\beta^*_{China} = 0.017$, $\beta^*_{Finland} = 0.022$, and $\beta^*_{South\ Africa} = 0.011$.

**Table 3. Actual and optimal income distributions in four countries.**

| Country | Income | Lowest quintile | Second quintile | Third quintile | Fourth quintile | Highest quintile | Gini coefficient |
|---|---|---|---|---|---|---|---|
| **U.S.A.** | Actual | 3.1 | 8.3 | 14.1 | 22.7 | 51.9 | 0.45 |
| | Optimal ($\beta^* = 0.017$) | 14.3 | 15.6 | 17.3 | 20.0 | 32.8 | 0.17 |
| | Difference | -11.2 | -7.3 | -3.2 | 2.7 | 19.1 | 0.28 |
| **China** | Actual | 6.5 | 10.7 | 15.3 | 22.2 | 45.3 | 0.36 |
| | Optimal ($\beta^* = 0.017$) | 15.4 | 16.6 | 17.9 | 20.2 | 29.9 | 0.13 |
| | Difference | -8.9 | -5.9 | -2.6 | 2 | 15.4 | 0.23 |
| **Finland** | Actual | 9.4 | 14 | 17.4 | 22.3 | 36.9 | 0.25 |
| | Optimal ($\beta^* = 0.022$) | 15.5 | 17.1 | 18.5 | 20.6 | 28.4 | 0.12 |
| | Difference | -6.1 | -3.1 | -1.1 | 1.7 | 8.5 | 0.13 |
| **South Africa** | Actual | 2.4 | 4.8 | 8.2 | 16.5 | 68.2 | 0.57 |
| | Optimal ($\beta^* = 0.011$) | 15.8 | 16.2 | 16.9 | 18.5 | 32.6 | 0.14 |
| | Difference | -13.4 | -11.4 | -8.7 | -2 | 35.6 | 0.43 |

The optimal *β* value was calculated from $\frac{\partial W}{\partial \beta}\big|_{\beta^*} = 0$.

Difference = Actual income distribution–Optimal income distribution.

The universal feasible equality line was further investigated in the evolution of the household income dispersions in China from 1990 to 2016 (S1 Table). From the income dispersions, the $L$, $H$, $\mu$, and $\alpha$ values in the sigmoid individual welfare functions were calculated (S2 Table). Then, the optimal income distributions were evaluated by determining the social-welfare-maximizing $\beta$ values in the Boltzmann income distribution (S3 Table and S1 Fig).

Fig 7 shows the Lorenz curves for the actual and the optimal income distributions in China from 1990 to 2016. The Lorenz curves for the actual income distributions are widely dispersed, whereas the Lorenz curves for the optimal income distributions, or the feasible income equality lines, are much narrowly dispersed in between the diagonal (idealistic perfect equal) and the actual income lines.

From the Lorenz curves, the Gini coefficients were calculated (S3 Table), and the evolution of the Gini coefficients is plotted in Fig 8. The Gini coefficient for the actual income distributions shows noticeable changes with trackable trends: it increases from 0.30 (1990) to 0.40 (2010), and then decreases slightly down to 0.36 (2016). On the other hand, the Gini coefficient for the optimal income distributions presents almost a flat line with little variations between 0.12 (1990) and 0.15 (2016). This result suggests that a universal feasible equality line could be time-independent, therefore serving as a reference over a long period of time.

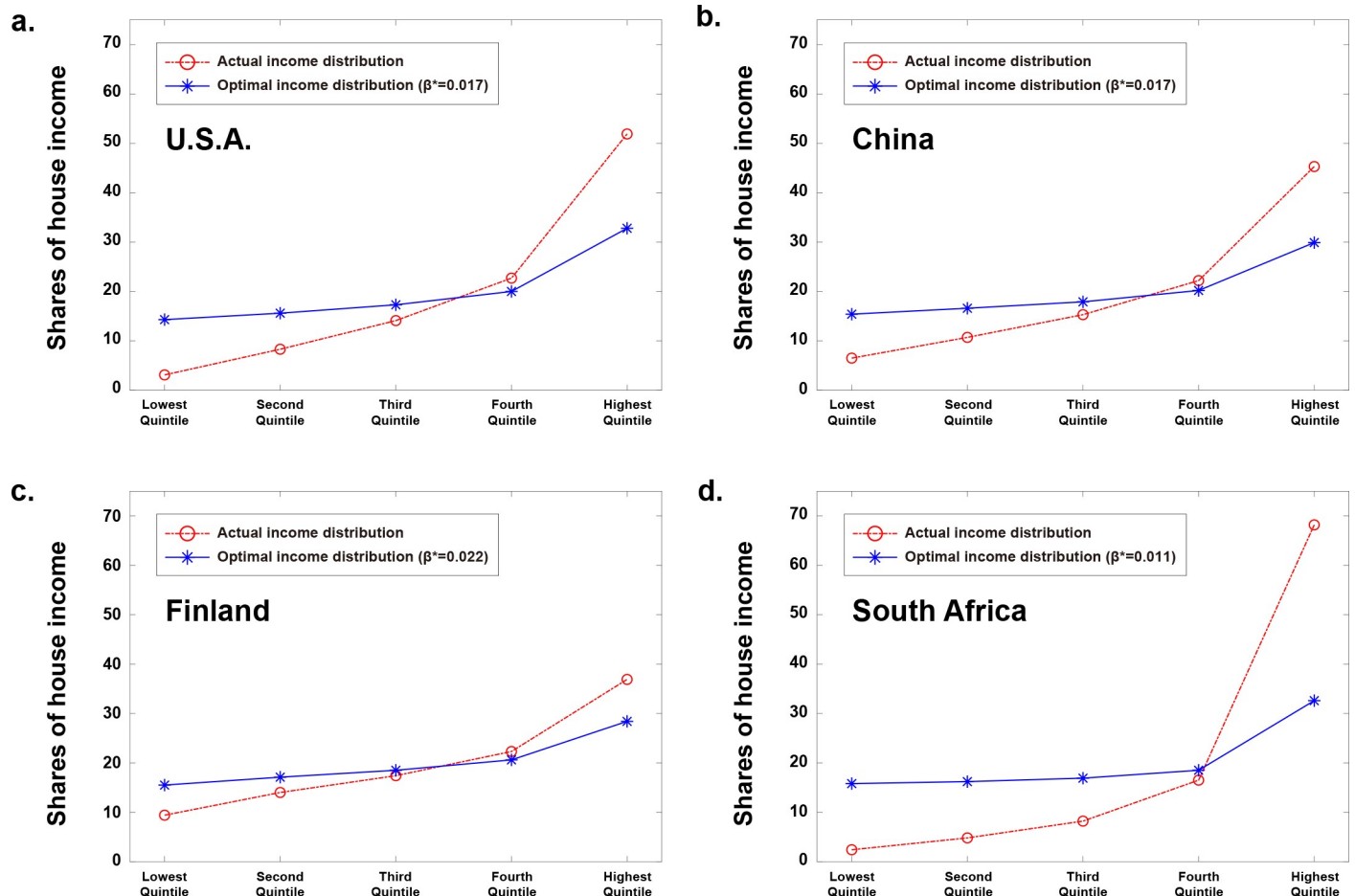

**Fig 5. Actual and optimal income distributions in four countries.** Compared to the actual income distribution, the optimal income distribution shows higher income in the lower quintiles and less income in the highest quintile.

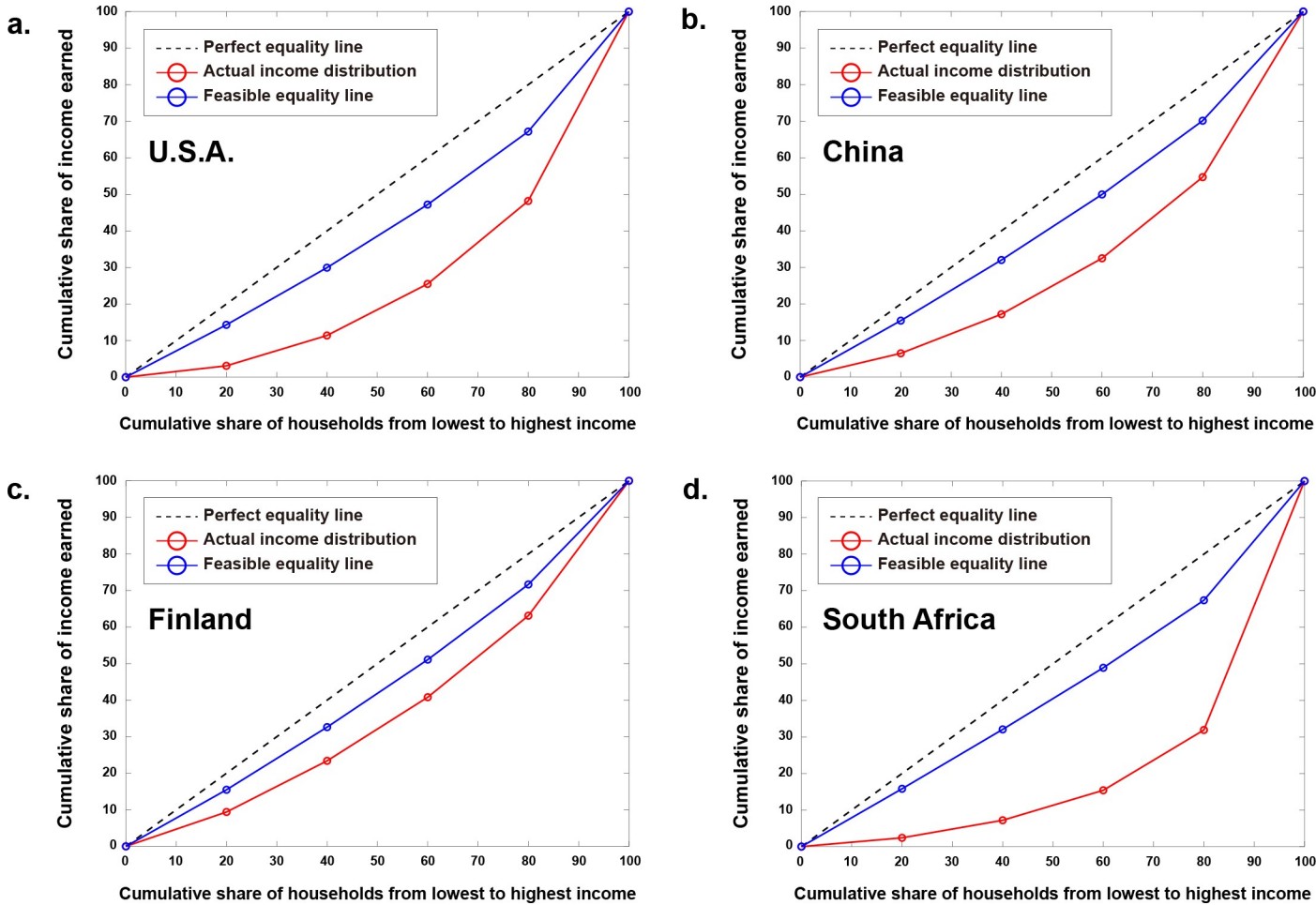

**Fig 6. Lorenz curves for the actual and the optimal income distributions in four countries.** The corresponding Gini coefficients can be found in Table 3. Note that the feasible equality lines (blue) are similar in all countries.

## Discussions

Over the past decades, income inequality has been severely worsened worldwide [50, 51]. Some scholars recently argue that the US income inequality has increased less than previously thought due to a more modest increase of capital income at the top [15, 52, 53]. However, according to Emmanuel Saez at UC Berkeley, in 2018, America's top 10 percent average more than nine times as much income as the bottom 90 percent, and America's top 1 percent average over 39 times more income than the bottom 90 percent. In addition, the super richest 0.1 percent take in 196 times as much income as the bottom 90 percent [15]. Thus, the U.S.A. currently experiences significant income inequality. One of the reasons for the current US income inequality might be the emergence of neoliberalism. Thomas Piketty (2014) argued that worsening income inequality is an unavoidable outcome of free-market capitalism and that neoliberalism leads to greater income inequality due to the limit of government regulations [13]. If government regulations are essential to reverse the income inequality, and then the feasible income equality line described in this study can be a useful and practical guideline. If the actual income distribution is far away from the feasible equality line as in the empirical analysis, these countries might need government interventions to redistribute income.

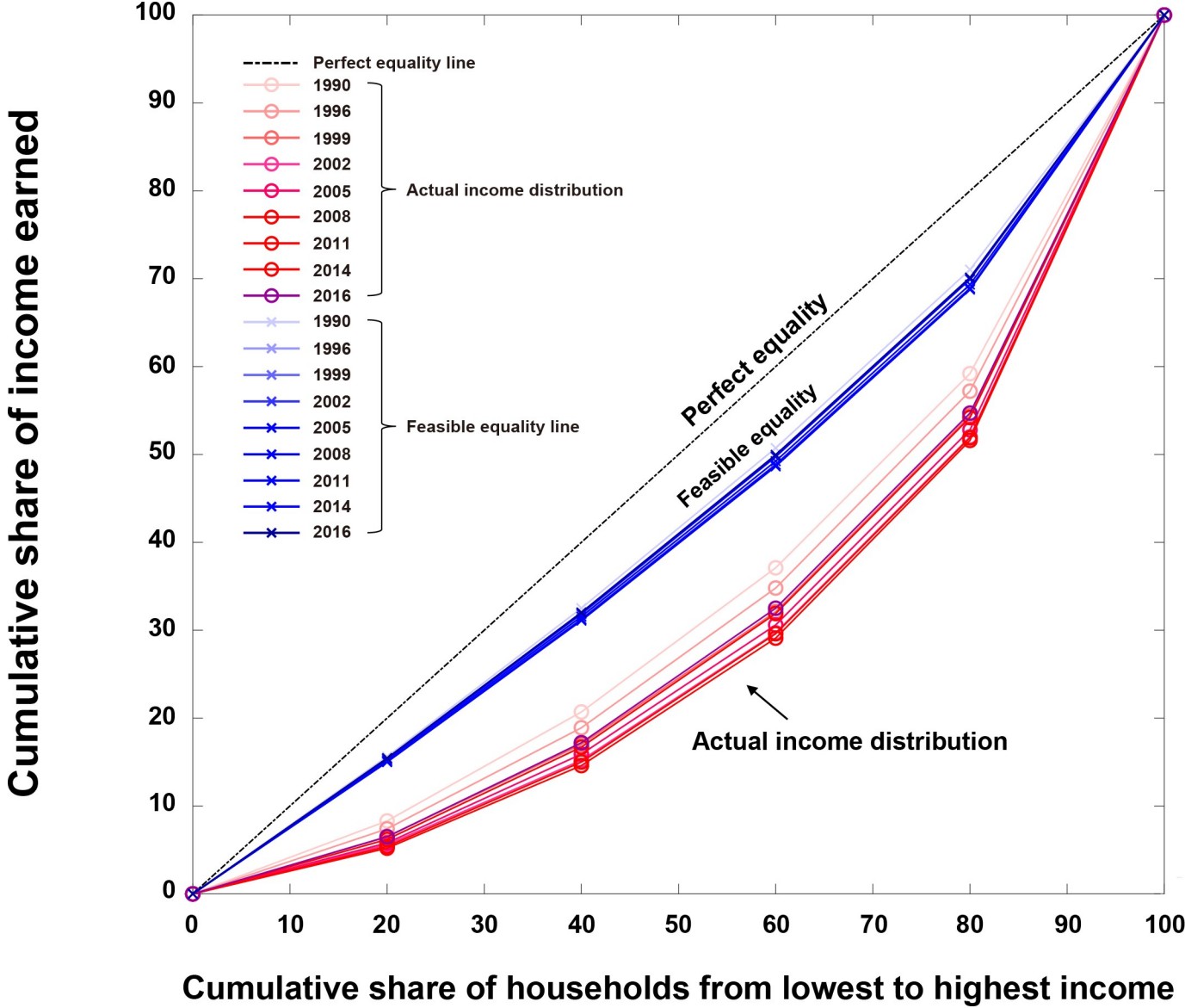

**Fig 7. Lorenz curves for the actual and the optimal income distributions in China from 1990 to 2016.** The corresponding Gini coefficients can be found in S3 Table. Note that the feasible equality lines (blue) are similar, while the actual income distributions are widely dispersed over time.

It should be noted that the feasible income equality line in this study can be further fine-tuned with detailed data. As proof of principle, we considered only five subgroups in a country in the empirical data analysis. However, our model can be easily extended to a country having finely divided subgroups. Further work is needed to find the reasonable critical low-income ($L$) and the critical high-income ($H$) values in the sigmoid individual welfare function for the finely subgrouped countries. In addition, in our empirical data analysis, as proof of principle, the income distribution factor was assumed to be proportional to the actual income. However, the income distribution factor is a measure of economic contributions and should be quantitatively determined by considering various factors such as intelligence, personalities, and physical and social skills. Therefore, it is essential to develop a more rigorous and reasonable formulation of the income distribution factor ($\tilde{E}_i$) in the Boltzmann income distribution.

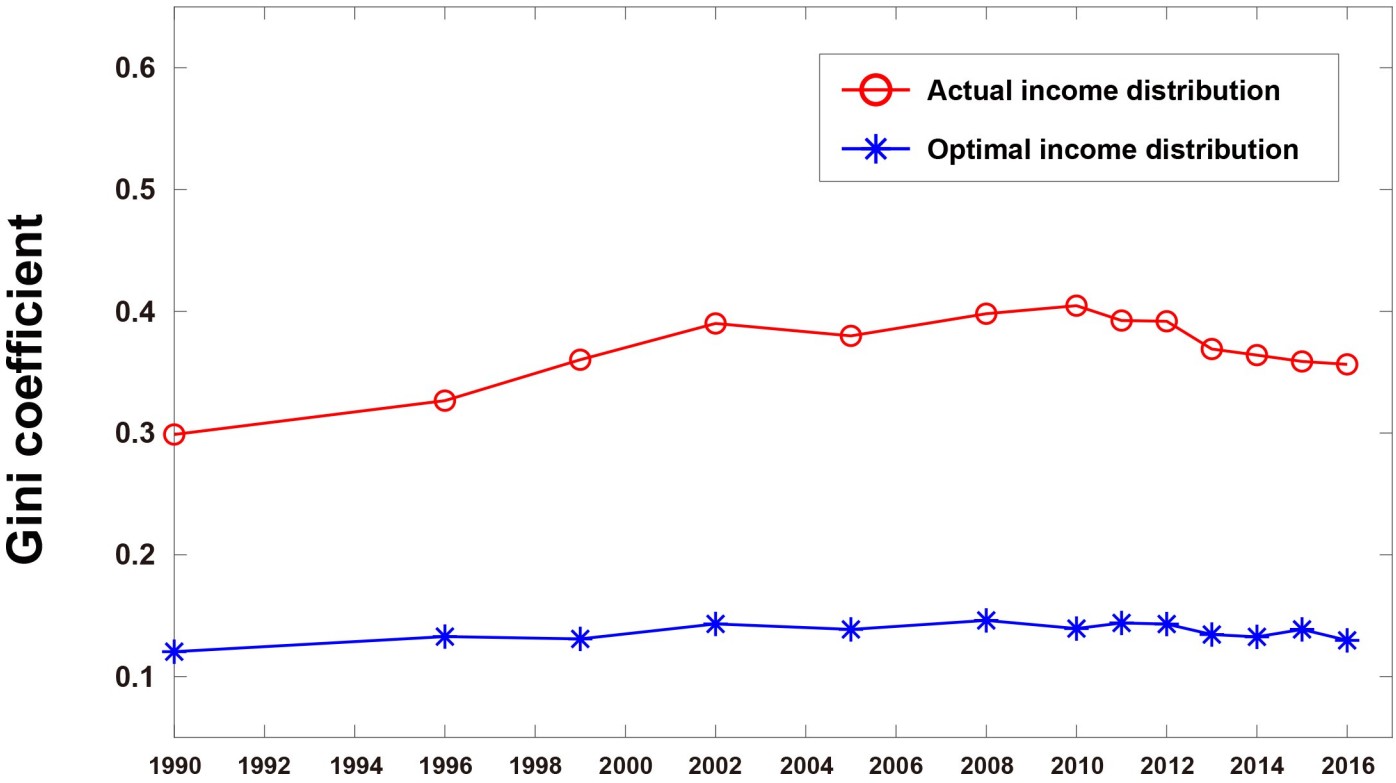

**Fig 8. Evolution of Gini coefficients in China from 1990 to 2016.** The Gini coefficient values at the data points can be found in S3 Table.

## Conclusions

In conclusion, we demonstrated that an optimal income distribution representing feasible income equality could be modeled using the sigmoid welfare function and the Boltzmann income distribution. In the empirical data analysis, we then showed how the optimal income distribution could be evaluated in the four countries. We revealed that the feasible income equality line could be time-independent and universally applicable to multiple countries. We believe that our work can be used as direct input for future theoretical and empirical studies on income inequality or government policies, which we anticipate could open a new window to feasible equality in the real world.

## Supporting information

**S1 Fig. Total social welfare as a function of $\beta$ value in China from 1990 to 2016.** The social welfare plots are normalized to the number of individual households in each subgroup (0.2**N**, where **N** is the total number of individual households in China). The total social welfare functions of China from 1990 to 2016 are maximized at $\beta^*_{China} = 0.016 \sim 0.019$.
(DOCX)

**S1 Table. Share of household income in China from 1990 to 2016.**
(DOCX)

**S2 Table. Parameter values for the annual social welfare functions in China from 1990 to 2016.**
(DOCX)

**S3 Table. Actual and optimal income distributions in China from 1990 to 2016.**
(DOCX)

## Acknowledgments

The authors sincerely thank professors Timothy D. Mount and Kieran P. Donaghy at Cornell University for their thoughtful comments and encouragements.

## Author Contributions

**Conceptualization:** Ji-Won Park, Chae Un Kim.

**Data curation:** Ji-Won Park.

**Formal analysis:** Ji-Won Park.

**Methodology:** Ji-Won Park, Chae Un Kim.

**Project administration:** Ji-Won Park.

**Visualization:** Ji-Won Park.

**Writing – original draft:** Ji-Won Park, Chae Un Kim.

**Writing – review & editing:** Ji-Won Park, Chae Un Kim.

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
