## [Decision Letter · Decision Letter 0]

27 Jan 2021

PONE-D-20-36244

Getting to a feasible income equality

PLOS ONE

Dear Dr. Park,

Thank you for submitting your manuscript to PLOS ONE. After careful consideration, we feel that it has merit but does not fully meet PLOS ONE’s publication criteria as it currently stands. Therefore, we invite you to submit a revised version of the manuscript that addresses the points raised during the review process.

As you may see from their reports, our two referees consider your paper enough interesting, important  and novel to be published in PLOS ONE. I fully share their opinion. You also will find in these reports some minor comments and suggestions asking you clarify some concepts or include some additional information, that in my opinion are pertinent and easily you can answer and include in a revised version of your  manuscript.

Please. let me here apologize for the delay in sending this decision.

We look forward to receiving your revised manuscript.

Kind regards,

Alejandro Raul Hernandez Montoya, Ph D

Academic Editor

PLOS ONE

Reviewers' comments:

Reviewer's Responses to Questions

**Comments to the Author**

1. Is the manuscript technically sound, and do the data support the conclusions?

Reviewer #1: Yes

Reviewer #2: Yes

2. Has the statistical analysis been performed appropriately and rigorously? 

Reviewer #1: Yes

Reviewer #2: Yes

3. Have the authors made all data underlying the findings in their manuscript fully available?

Reviewer #1: Yes

Reviewer #2: Yes

4. Is the manuscript presented in an intelligible fashion and written in standard English?

Reviewer #1: Yes

Reviewer #2: Yes

5. Review Comments to the Author

Reviewer #1: The subject of the paper is appealing. We need to have good indicators to measure income inequality because the gap between rich and poor has been unprecedentedly widening. The authors point out that the Gini coefficient has been a prevailing indicator but need an alternative measure that is realistic and feasible. I agree. I understand the intention of the argument from page 1 to page 8.

However, it is getting hard to understand from page 9. They need to clarify what is "income allocation preference." Does this mean the "sum of various talents" of the individuals? If so, It sounds like justifying the idea that talented people should be affluent in proportion to their talents. (Maybe I'm wrong)

The authors wrote that income should be distributed based on the income allocation preference. I am not still convinced of this argument, but I put it aside for a while, waiting for the author's reply.

The next question; why do they choose the Bolzman distribution an appropriate measure? How does this model connect to the idea that that income should be distributed based on the income allocation preference?

Although it remains some questions above, the calculation results can be an alternative indicator that we feel feasible.

In conclusion, this paper is worth publishing with minor modifications.

Reviewer #2: This is an interesting proposal to improve the Gini coefficient as a better measure of income inequality of nations. I consider that this work could be of interest to a broad community of social scientists and governmental policy makers.

The authors have modeled "feasible" (or optimal) income distributions that could be implemented in the calculation of new Gini coefficients of nations. They have used the sigmoid welfare function and the Boltzmann income distribution to generate such optimal income distributions and then used to calculate new Gini coefficients of four nations (USA, China, Finland and South Africa). As a consequence, the new Gini coefficients are now narrowly distributed. The present proposal involves a rather technical methodology to determine the respective optimal income distributions of each country. For this reason, in order to get a better in site of the proposal, I suggest that the authors should extend their excise shown in Table 3 for one year,

to a five-year window. In this way it will be easier to understand the evolution of the difference between the

calculation of the two versions of each Gini coefficient. In particular, it will be interesting to learn how China's

Gini coefficient has evolved in the last five years.

6. PLOS authors have the option to publish the peer review history of their article (what does this mean?). If published, this will include your full peer review and any attached files.

Reviewer #1: No

Reviewer #2: No

---

## [Author Response · Author response to Decision Letter 0]

15 Feb 2021

Please see the uploaded 'Response to Reviewers'.

---

## [Decision Letter · Decision Letter 1]

15 Mar 2021

Getting to a feasible income equality

PONE-D-20-36244R1

Dear Dr. Park,

We’re pleased to inform you that your manuscript has been judged scientifically suitable for publication and will be formally accepted for publication once it meets all outstanding technical requirements.

Kind regards,

Alejandro Raul Hernandez Montoya, Ph D

Academic Editor

PLOS ONE

Additional Editor Comments (optional):

Reviewers' comments:

Reviewer's Responses to Questions

**Comments to the Author**

1. If the authors have adequately addressed your comments raised in a previous round of review and you feel that this manuscript is now acceptable for publication, you may indicate that here to bypass the “Comments to the Author” section, enter your conflict of interest statement in the “Confidential to Editor” section, and submit your "Accept" recommendation.

Reviewer #1: All comments have been addressed

Reviewer #2: All comments have been addressed

2. Is the manuscript technically sound, and do the data support the conclusions?

Reviewer #1: Yes

Reviewer #2: (No Response)

3. Has the statistical analysis been performed appropriately and rigorously? 

Reviewer #1: Yes

Reviewer #2: (No Response)

4. Have the authors made all data underlying the findings in their manuscript fully available?

Reviewer #1: Yes

Reviewer #2: (No Response)

5. Is the manuscript presented in an intelligible fashion and written in standard English?

Reviewer #1: Yes

Reviewer #2: (No Response)

6. Review Comments to the Author

Reviewer #1: I read your revised version carefully. I think you addressed my questions adequately. As long as I am an economist, I feel this paper is quite interesting and worth publishing. But due to the lack of my knowledge of physics, I am not 100% sure whether it is relevant to apply the Boltzman function to this phenomenon.

Therefore, I accept the paper, but I would like to leave the final decision to another judge and the editor.

Reviewer #2: (No Response)

7. PLOS authors have the option to publish the peer review history of their article (what does this mean?). If published, this will include your full peer review and any attached files.

Reviewer #1: No

Reviewer #2: No

---

## [Editor Report · Acceptance letter]

19 Mar 2021

PONE-D-20-36244R1 

Getting to a feasible income equality 

Dear Dr. Park:

I'm pleased to inform you that your manuscript has been deemed suitable for publication in PLOS ONE. Congratulations! Your manuscript is now with our production department. 

Kind regards, 

on behalf of

Dr. Alejandro Raul Hernandez Montoya 

Academic Editor

PLOS ONE